# The Characteristics of Benzodiazepine Prescribing in the Republic of Srpska, Bosnia and Herzegovina

**DOI:** 10.3390/medicina58080980

**Published:** 2022-07-22

**Authors:** Žana M. Maksimović, Mladen Stajić, Miloš P. Stojiljković, Svjetlana Stoisavljević Šatara, Nataša Stojaković, Ranko Škrbić

**Affiliations:** 1Emergency Department, Primary Healthcare Centre Modriča, Svetosavska 15, 74480 Modriča, Bosnia and Herzegovina; mladen_s88@hotmail.com; 2Centre for Biomedical Research, Faculty of Medicine, University of Banja Luka, Save Mrkalja 14, 78000 Banja Luka, Bosnia and Herzegovina; milos.stojiljkovic@med.unibl.org (M.P.S.); ranko.skrbic@med.unibl.org (R.Š.); 3Department of Pharmacology, Toxicology and Clinical Pharmacology, Faculty of Medicine, University of Banja Luka, Save Mrkalja 14, 78000 Banja Luka, Bosnia and Herzegovina; cecasat@yahoo.com (S.S.Š.); natasa.stojakovic@med.unibl.org (N.S.)

**Keywords:** benzodiazepines, primary healthcare, inappropriate prescribing, long-term use

## Abstract

*Background and Objectives*: Benzodiazepines (BZDs) are among the most prescribed psychotropic drugs and significant number of patients use these drugs for longer periods than recommended. The objective of this study was to determine the factors associated with prescribing of BZDs at the primary healthcare level. *Materials and Methods*: A retrospective analysis of family physicians’ prescriptions from the databases of family medicine teams of the Republic of Srpska was performed. The number of BZDs users, as well as the total number of prescriptions, were determined. Thereafter, it was determined which specific BZD had been prescribed, in which dose, for how long, as well as the specific social and demographic characteristics of patients to whom the drugs were prescribed. *Results*: The results showed that 38.47% of patients used the BZDs for a period longer than six months. The most frequent BZDs prescribed were the intermediate-acting BZDs, primarily bromazepam (58.69%). Two thirds of patients were women. The average age of the patients was 60, 60.46% of patients were single, and 69.68% lived in urban areas. The longer uses of BZDs were recorded in women, the elderly, single people and those who lived in urban areas, while higher doses of BZDs were prescribed to men, as well as younger and married people. The highest positive correlation was found between the dose and length of use of BZD. *Conclusions*: A significant percentage of patients used BZDs for a time period longer than recommended. Caution is necessary when prescribing BZDs to women, the elderly, patients that live in urban areas and patients who are single. When prescribing BZDs, family physicians should be aware of their potential interactions and addictive potentials.

## 1. Introduction

Benzodiazepines (BZDs) are GABA-A agonists that exert anxiolytic, sedative, hypnotic, anticonvulsant and myorelaxant effect via this mechanism [1]. These are among the most often prescribed psychotropic drugs. Nowadays, there are four indications for the rational use of BZDs in patients with psychiatric diagnoses, namely: panic disorder, generalised anxiety disorder, social phobia, and insomnia [2]. The latest guidelines limit the use of BZDs in these conditions to up to 4–12 weeks, while any prolonged use is considered unjustified [3,4]. In addition to these disorders, BZDs are justifiably used as anticonvulsants, especially in emergencies. Despite the well-defined recommendations, BZDs are often prescribed as “off-label” drugs for clinical conditions not confirmed by clinical trials and for longer periods of time than recommended. Some examples are schizophrenia and depression, in which BZDs are often used as continuous therapy. Recommendations are that in these conditions BZDs are used only in case of agitation. A study conducted in the Republic of Srpska found that as many as a quarter of BZDs were prescribed for non-psychiatric diagnoses such as musculoskeletal diseases, followed by epilepsy, hypertension, arrhythmias, and chest pain [5]. Maric et al. found that, in South-Eastern Europe, as many as 81.9% of patients were discharged from psychiatric clinics with a high dose of BZDs (equivalent to 5 mg lorazepam) [6].

The use of BZDs in the elderly population is associated with significant risks, such as psychological drug dependence, impaired judgment, confusion, reduced ability to drive, and frequent falls [4,7]. Almost 28% of all falls among people over the age of 80 are related to BZD abuse. It has been found that the use of BZDs is associated with fatal falls in 9% of elderly individuals [8]. BZDs are even related to the increase in all-cause mortality [9]. Most studies have found correlations between older age, urban environment, single people, lower income, poorer education, and the long-term use of BZDs [10,11,12].

The main objective of this study was to determine the frequency, average dose, and length of use of BZDs in primary healthcare patients and to analyse the predictive factors for their irrational use.

## 2. Materials and Methods

This was a retrospective cross-sectional study performed in adult patients to whom the BZDs were prescribed by family physicians in the Republic of Srpska. The Republic of Srpska, with a total population of 1.2 million, represents one of two constitutive entities of Bosnia and Herzegovina. The health system in the Republic of Srpska is centralised, with planning, regulation, and management functions held by the Ministry of Health and Social Welfare. The Health Insurance Fund (HIF) provides universal health insurance coverage for the population and operates on the basis of solidarity and mutuality. It is the only body legally responsible for the collection and allocation of financial contributions to healthcare providers. The HIF reimburses prescribed medications as well, based on a list of refundable drugs, known as the “positive list”. The drugs are listed under international non-proprietary names (INN-ATC level 5) based on clinical guideline recommendations. The drugs on the positive list are dispensed in pharmacies as prescription drugs only.

### 2.1. Data Collection and Analyses

Data of prescribed drugs were collected form *WebMedic* database of primary healthcare of the Republic of Srpska. The *WebMedic* is an information system primarily designed for primary healthcare system and used by family practitioners in their daily practice. All BZDs prescriptions from February 2014 to February 2021 were collected from the database using the specific ATC codes for these drugs (N05BA, N05CD, and N03AE01) [13]. According to the Law on Pharmacy of the Republic of Srpska, one prescription can cover a maximum of one month’s therapy [14].

To preserve the anonymity of patients, every patient was coded and no name, initials, or exact date of birth (only the year of birth) were used. Basic socio-demographic data such as: gender (male/female), age (years), place of residence (urban area: municipality population ≥ 20,000 inhabitants/rural area: municipality population < 20,000 inhabitants), and marital status (married/single) were recorded. Additionally, the generic names of BZDs, with their ATC codes, the daily doses, and lengths of use (number of months in one year), were extracted from each individual prescription.

This study was based on anonymous data analyses and did not involve contacts or any intervention with patients. Therefore, it was not necessary to obtain permission from the Ethics Committee.

To compare the doses of different BZDs, the equivalences of doses were devised using the table of bioequivalence [13,15]. The equivalent dose (ED) matches clinical equivalence when a patient is “switched” from one BZD to another; ED = 1 equals 10 mg diazepam and equivalent doses of other BZDs (Table 1). Furthermore, for easier comparison, the elimination half-life (t_1/2_) of BZDs were used for drug grouping: short-acting BZDs (t_1/2_ < 12 h), intermediate-acting BZDs (t_1/2_ = 12–24 h), and long-acting BZDs (t_1/2_ > 24 h) (Table 1) [15,16]. A significant number of patients used different BZDs over the observed period of time, sometimes from the same group, sometimes from a different group, and those patients were put in a separate, fourth category.

### 2.2. Statistical Analyses

The Kolmogorov-Smirnov test was used and a significant deviation from normal distribution of continual data was shown, which imposed a need for the implementation of nonparametric statistical tests. Descriptive statistics consisted of expressing the data as a mean with its standard deviation (SD) and a 95% confidence interval (CI). Patients who lacked data on a specific parameter were not included in the analysis. A chi-square test was used to analyse the connection between specific categorical variables. Continuous variables were analysed by the Mann-Whitney U-test or the Kruskal-Wallis test. Multiple regression analysis has been conducted to analyse the impact of different parameters on the main variables: the length of use and the dose. The statistical significance level was set at *p* < 0.05. Data were processed using IBM SPSS 18.0 software.

## 3. Results

The total number of BZD prescriptions during a 7 year period was 1,125,632, which represented 2.98% of all prescriptions made by family practitioners in the same period. That number was prescribed to 151,204 different patients (10.63% of all patients).

Two-thirds of patients with BZD prescriptions were women, the average age of patients was 60.46 ± 15.00, 60% of patients were single, and two-thirds of BZDs users lived in urban areas (Table 2).

### 3.1. Type of BZDs Prescribed

A total of 98.67% of all prescribed BZDs corresponded to five BZDs: bromazepam 54.71%, diazepam 17.80%, alprazolam 16.80%, clonazepam 5.51%, and lorazepam 3.85%. Most prescribed BZDs, by length of their effect, were intermediate-acting BZDs (660,589 prescriptions or 58.69%), from which bromazepam comprised 93.22% and was the most prescribed BZD in the Republic of Srpska. In addition to that, lorazepam made up to 6.57% of the prescribed intermediate-acting BZDs. Subsequently, there were the long-acting BZDs (273,648 prescriptions or 24.31%), out of which 73.21% was diazepam and 22.67% was clonazepam. The least prescribed were the short-acting BZDs (191,386 prescriptions or 17.00%) and almost exclusively alprazolam (98.80%). The basic data of the type of BZDs, dosage, and length of use for BZDs are shown in Table 3.

The short-acting BZDs were more often prescribed to younger patients, while the long-acting BZDs were prescribed to older patients (χ^2^ = 14.72, *p* = 0.022).

A significant percentage of patients used several different BZDs, and more than half of those patients used BZDs continuously. In relation to type of BZDs by length of effect, short-acting BZDs were taken over the longest period of time (Kruskal-Wallis test: χ² = 6535.53, *p* < 0.001). (Figure 1).

Patients that combined several different BZDs used higher doses. Furthermore, people taking short-acting BZDs used higher doses (χ² = 10605.94, *p* < 0.001) while people taking intermediate-acting BZDs used the lowest doses (Figure 2). Out of the five most prescribed BZDs, bromazepam, as the most prescribed BZD, had the lowest ED of 0.69. The ED of diazepam amounted to 0.83, alprazolam to 1.50, clonazepam to 3.87, and lorazepam to 2.98.

### 3.2. Dose of BZDs Prescribed

The average dose per patient amounted to 0.88 ED (Table 3). The higher doses were more often prescribed to male patients (U = 6.13 × 10^8^, *p* < 0.001), to those who were younger (χ^2^ = 64.63, *p* < 0.001), single (U = 6.72 × 10^8^, *p* < 0.001), and to patients living in rural areas (U = 5.09 × 10^8^, *p* < 0.001).

To analyse the influence of different parameters on the dose of the BZD used, the multiple regression analysis (MRA) was conducted (Table 4). It showed a general statistical significance (F = 562.279, *p* < 0.001), with certain parameters having different effects. Those patients who used BZDs for longer period of time (impact 23.5%), younger patients (impact 9.4%), men (impact 8.6%), and single people (impact 2.4%) used higher doses.

### 3.3. Length of Use of BZDs

Half of patients used BZDs during the recommended 3-month period, while 30% used BZDs for longer than 10 months (average 5.5 months; Table 3). Women used BZDs for longer periods of time (U = 5.81 × 10^8^, *p* < 0.001), as well as the older patients (χ^2^ = 2576.66, *p* < 0.001), married people (U = 6.73 × 10^8^, *p* < 0.001), and patients living in urban areas (U = 4.95 × 10^8^, *p* < 0.001).

To analyse the influence of different parameters on length of use of BZDs, the MRA was conducted (Table 5). It showed a general statistical significance (F = 772.781, *p* < 0.001) of the length of use, with different impacts of certain parameters. A longer length of use was mostly influenced by a higher dose (impact 22.9%) and age (older patients had longer length of use; impact 18.7%). Being a woman (impact 4.5%) or married (impact 1.8%) also influenced longer length of use.

The strongest relation was found between the dose and the duration of BZD use (Figure 3). Patients that were taking highest doses used BZDs for longest time period, and vice versa.

## 4. Discussion

The results of this study have shown that almost 3% of all prescriptions made by family physicians belong to BZDs, and 10% of all primary healthcare patients used BZDs. Study by Maric et al. found that, at discharge from hospitals in Croatia, Macedonia and Serbia, 81.9% of patients had benzodiazepines prescribed. They found that the factors associated with the prescriptions were exclusively clinical factors, while sociodemographic factors were not found to influence the benzodiazepine prescriptions at discharge [6]. The BZD users were more often women, elderly, single, and those living in urban areas. Gender-wise, women took BZDs almost twice as often as men, which concurs with the results from other studies [17,18,19]. Women took smaller doses (ED < 1) compared to men, who mostly took doses between 1 and 2 ED. Women were prescribed BZDs for longer periods. The correlation between the long-term use of BZDs and gender vary from study to study, mostly concurring with the current results, but there have also been some contradictory findings [19,20].

Regarding the age of patients prescribed with BZD, patients older than 40 were predominant, with the percentage of elderly patients (older than 65) being 39.48%. This is in agreement with the results from other countries, that found BZDs were mainly used by older patients, and for longer periods of time than in younger patients [17,18,19,20,21,22]. This is probably due to a greater risk of depression and anxiety in the elderly population, as well as an increased incidence of organic and non-organic sleep disorders. Elderly patients used long-acting BZDs more often, which contrasts with the recommendations and an increased risk of accumulation of toxic doses of BZDs, due to the fact that elderly people have reduced glomerular filtration rates. The present results are in accordance with the results of other studies, which classifies BZDs, and specifically long-acting BZDs, as among the top five most inappropriately prescribed drugs in the elderly population [23]. Elderly patients used smaller doses, which is logical, based on the narrower therapeutic index (lower creatinine clearance values, reduced liver function, polypharmacy, interaction with other drugs, smaller body weight), and coincides with global data [17,18,20]. The use of higher doses of BZDs is associated with languor, lethargy, withdrawal from social interactions, and frequent falls in the elderly. This gives the impression that doctors showed some precaution when prescribing anxiolytics and sedatives to elderly patients, since minimal doses were usually prescribed.

The BZD users were more often single people (nearly 60%), and other studies showed similar results [12,21]. This study showed that married people used BZDs for longer period. This is probably the consequence of a general trend of getting married at an older age and is in accordance with the notion that older people use anxiolytics and hypnotics for longer periods. Furthermore, married people take smaller doses compared to single people. Since the MRA found only a small effect of marital status on the BZD dose and length of their use, it is reasonable to assume that the results, to a significant extent, correspond to the age of single patients.

Patients from urban areas used BZDs more often (70:30%). Romans et al. showed that mood disorders were more often seen in patients from urban areas compared to rural areas [24]. Moreover, patients in urban areas take BZDs for longer periods of time [24]. This is expected, taking in consideration the faster lifestyle, noise, traffic, air pollution, and other factors to which inhabitants of urban areas are more exposed. The citizens of rural areas took slightly higher doses. A potential explanation is the distance and inaccessibility of health services in rural areas. Physicians often observe that patients who live in rural regions demand prescriptions for longer periods of time due to their long distance from their family physicians. Since the prescription is issued exclusively for monthly therapy, doctors may do a “favour” to the patient by putting a higher dose or frequency of medication on the prescription than the one used by the patient. In this way, the patient can pick up a larger number of packages at a pharmacy and, thus, reduce the frequency of visits to the doctor. Although the difference was statistically significant, it is clinically irrelevant (0.87 vs. 0.92 ED). A study by Tahiri et al., in Kosovo, found a correlation between the inappropriate BZD use and older age, a middle education level, and rural environments [25].

For better comparability of data, it is recommended that the long-term use in studies covers a period longer than 6 months [10]. Data showed that almost 40% of patients used BZDs for longer than 6 months, and about a quarter of them used BZDs as a continuous treatment. Prescribing guidelines strongly emphasise that BZDs should only be prescribed for up to 12 weeks, including a discontinuation period. In practice, these drugs are prescribed for much longer periods [26]. This could be explained by the over-reliance of patients with sleeping disorders on these drugs, but also by the fact that BZDs form a psychological dependence and patients quickly habituate to their dose. It is unclear why doctors do not intervene and replace BZDs with other more suitable drugs. On the one hand, there is a significant pressure on doctors to continue to prescribe the drugs that had already brought a calming effect to their patients. On the other hand, a patient will often reject antidepressants because of social stigma and personal prejudice towards people who take antidepressants.

The studies have shown that family physicians are most often familiar with suggestions and guidelines for prescribing BZDs, but they are found in an ambivalent position between the recommendations and the needs of the patient and insecurities of how to solve the patient’s problems and maintain a good doctor-patient relationship [27,28]. Furthermore, there is a resistance on behalf of doctors to introduce antidepressants even if there is a clear indication to do so, potentially due to the fear of antidepressant side effects when not titred properly [26]. Some authors, primarily clinicians, insist that a small percentage of long-term BZDs users have a favourable risk-benefit ratio and relatively fewer adverse effects [29]. They believe that the risk-benefit ratio with long-term use of BZDs has not been properly studied compared to the alternative pharmacotherapeutic approaches, such as selective serotonin reuptake inhibitors (SSRIs) [30].

Patients prescribed with the highest doses of BZDs took them for the longest periods. This confirms the possible addictive potential of BZDs. Initially, lower doses of BZDs give a patient a calming effect, which does not occur after prolonged use, and the patient reaches for a higher dose [31,32]. Particular attention should be paid to the “withdrawal” phenomenon of this class of drugs. It is estimated that up to 90% of patients experienced psychological and physical withdrawal symptoms after the discontinuation of BZDs [1,7]. Fear of discomfort after the discontinuation of the drug leads patients into continuous drug use [33]. Another suspicion, that longer use of BZDs leads to addiction, is supported by the findings, in that patients who used a combination of different BZDs are those ones who used them for the longest periods. It is assumed that by increasing their tolerance, the patient becomes unsatisfied with the effects of a given BZD and so, by hoping to acquire the same calming effect as at the start of therapy, a patient “switches” from one BZD to another.

When it comes to the type of BZDs, intermediate-acting BZDs were prescribed most frequently, followed by the long-acting BZDs, and with short-acting BZDs being the least prescribed. Bromazepam, which comprised 54.71% of all prescribed BZDs, was the most prescribed anxiolytic in this study. This is mostly due to the prescribing practice and the drug becoming the “go-to” drug among doctors [27]. We believe that there is no other more logical explanation for this excessive number of prescriptions for the abovementioned drug compared to, for example, lorazepam, which belongs to the same group. The influence of pharmaceutical companies and lobbying activities cannot be neglected. Studies from other countries serve as further examples, in which the data differ drastically [17,34]. Pharmacoeconomics and the registration of a specific drug in a country are some additional reasons for prescribing habits—physicians are more likely to prescribe drugs that have been in use for longer and that are cheaper.

An equivalence test was performed to obtain equivalent doses between BZDs, which then could be compared. For this purpose, the ED was used; ED 1 matches 10 mg of diazepam. The ED is primarily a clinical category and the equivalence test was performed based on recommended doses that the patient should use during the “switch” from one BZD to another. The ED matches the dose that patients use more realistically, compared to the defined daily dose (DDD), which is more of a statistical parameter [35]. More than half of the patients used doses of less than 1 ED. This shows that, despite the excessive prescription of BZDs, doctors try to prevent overdosing by prescribing smaller doses. Then again, after taking in consideration the fact that anxiolytics are mainly used by elderly people, it is logical and justified that doses smaller than 1 ED are being prescribed [4]. On the other hand, as many as 18% of patients used a dose of at least two times higher than 1 ED.

Based on the BZD type, short-acting BZDs were taken for the longest periods. This is logical, after taking in consideration the fact that that these drugs have the fastest addiction rates. Their anxiolytic effects are short-lasting and incomplete, so the patients increase their doses and durations of drug taking “on their own” [21]. Patients taking short-acting BZDs also took the highest doses. Other studies have also reported the addictive potential of alprazolam [36].

Limitations of the study: A major limitation of the present study is that it is a retrospective study, in which no certain cause/effect relationship can be established. It was not possible to determine the reason for the first introduction of BZD to these patients. There is a lack of data on potential attempts to stop using BZD or eventual advice from doctors about the addictive potential of BZD. Data were obtained exclusively through an electronic database. There are a few rural areas where prescriptions are still written by hand. Furthermore, the study refers to the number of issued prescriptions, without insight into the number of BZDs purchased from pharmacies.

## 5. Conclusions

It can be concluded that a significant number of patients used BZDs and that their prolonged use is quite common. There is a need to be cautious in the initial prescribing of BZDs, as even their short-term use can cause addiction due to their strong addictive potential. Connections with prolonged BZD use were found in women, the elderly, and married people. Additional attention should be paid when prescribing BZDs to those patients. The strongest positive correlation was found between the length of use and the dose, implying a potential habituation to BZDs. An attempt to reduce the prevalence of BZD use should be aimed for, especially in patient categories that have unjustified use of these drugs.

## Figures and Tables

**Figure 1 medicina-58-00980-f001:**
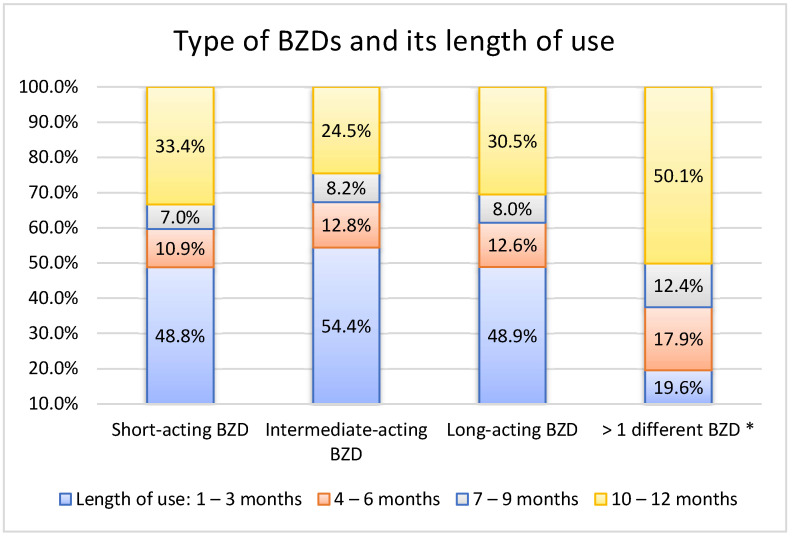
Length of use depending on the type of benzodiazepine (BZD). * A significant number of patients had used different types of BZDs during the surveyed time period.

**Figure 2 medicina-58-00980-f002:**
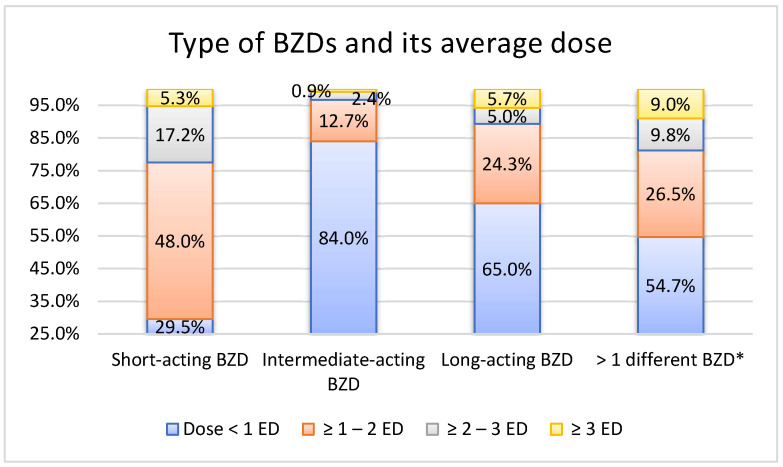
Average dose depending on type of benzodiazepine (BZD). * A significant number of patients had used different types of BZDs during the surveyed time period.

**Figure 3 medicina-58-00980-f003:**
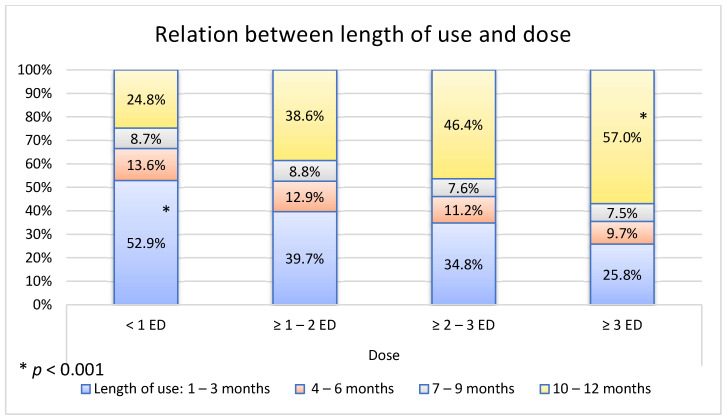
Relation between length of use and dose of benzodiazepines (BZDs).

**Table 1 medicina-58-00980-t001:** Duration of action and equivalent doses of benzodiazepines.

Duration of BZD Action	INN	ATC Code	ED
	Alprazolam	N05BA12	0.5 mg
Short-acting	Brotizolam	N05CD09	0.25 mg
(t_1/2_ < 12 h)	Midazolam	N05CD08	5 mg
	Oxazepam	N05BA04	20 mg
Intermediate-acting (t_1/2_ = 12–24 h)	Bromazepam	N05BA08	5 mg
	Lorazepam	N05BA06	1 mg
	Nitrazepam	N05CD02	10 mg
	Diazepam	N05BA01	10 mg
	Clorazepate	N05BA05	15 mg
Long-acting	Flurazepam	N05CD01	20 mg
(t_1/2_ > 24 h)	Clobazam	N05BA09	20 mg
	Clonazepam	N03AE01	0.5 mg
	Medazepam	N05BA03	10 mg
	Prazepam	N05BA11	10 mg

BZD: benzodiazepine; t_1/2_: elimination half-life; INN: international non-proprietary name; ATC code: the anatomical therapeutic chemical code; ED: equivalent dose.

**Table 2 medicina-58-00980-t002:** Average dose and length of use of benzodiazepines related to socio-demographic characteristics of patients.

Variable	N	%	ED	Length of Use
			Mean ± SD	Mean ± SD
**Gender**				
Male	52,254	34.56	0.98 ± 0.96	5.39 ± 4.51
Female	98,950	65.44	0.83 ± 0.79	5.72 ± 4.50
**Age**				
18–40 years	16,782	11.10	0.96 ± 1.03	3.68 ± 3.92
41–65 years	74,731	49.42	0.91 ± 0.89	5.47 ± 4.49
≥66 years	59,691	39.48	0.83 ± 0.75	6.27 ± 4.53
**Marital status**				
Married	59,781	39.54	0.86 ± 0.82	5.65 ± 4.51
Single	91,423	60.46	0.89 ± 0.88	5.54 ± 4.51
**Place of residence**				
Urban areas	105,365	69.68	0.87 ± 0.84	6.35 ± 4.53
Rural areas	45,839	30.32	0.92 ± 0.90	5.26 ± 4.46
**Total**	**151,204**	**100.00**	**0.88 ± 0.86**	**5.59 ± 4.51**

Length of use in months (in one year); ED: approximately equivalent dosage between benzodiazepines, with 1 corresponding to 10 mg of diazepam; N: number of patients; SD: standard deviation.

**Table 3 medicina-58-00980-t003:** Dose, length of use, and type of benzodiazepines distribution.

Variable	N	%
**Length of use (mean ± SD: 5.59 ± 4.51)**		
1–3 months	73,061	48.32
4–6 months	19,980	13.21
7–9 months	12,988	8.59
10–12 months	45,175	29.88
**Dose (mean ± SD: 0.88 ± 0.86)**		
<1 ED	107,040	70.79
1–2 ED	30,928	20.45
2–3 ED	8147	5.39
>4 ED	5089	3.37
**Type of BZD**		
Short-acting BZD	15,435	10.21
Intermediate-acting BZD	85,030	56.24
Long-acting BZD	25,742	17.02
>1 different BZD	24,997	16.53
Total	151,204	100.00

Length of use in months (in one year); BZD: benzodiazepine; ED: approximately equivalent dosage between benzodiazepines, with 1 corresponding to 10 mg of diazepam; N: number of patients; >1 different BZD: Significant number of patients has used different type of BZDs during surveyed time period.

**Table 4 medicina-58-00980-t004:** The influence of different parameters on dose of benzodiazepines.

Factors	MRA for Dose		
	β	t	*p*
Length of use	0.235	65.55	0.001
Age	−0.094	25.90	0.001
Gender (male)	0.086	24.20	0.001
Marital status (married)	−0.024	6.89	0.001

MRA: multiple regression analysis, β: standardised coefficient beta; general statistical significance (F = 562.279, *p* < 0.001).

**Table 5 medicina-58-00980-t005:** The influence of different parameters on length of use of benzodiazepines.

Factors	MRA for Length of Use		
	β	t	*p*
Dose	0.229	65.55	0.001
Age	0.187	53.15	0.001
Gender (male)	−0.045	12.78	0.001
Marital status (married)	0.018	5.19	0.001

MRA: multiple regression analysis, β: standardised coefficient beta; general statistical significance (F = 772.781, *p* < 0.001).

## Data Availability

Not applicable.

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
