# Peer review of "The Characteristics of Benzodiazepine Prescribing in the Republic of Srpska, Bosnia and Herzegovina"

_medicina, 2022, doi:10.3390/medicina58080980_

Round 1

Reviewer 1 Report

I find the present article to be of extreme important for the scientific filed.

Here are some suggestions for improvement:

1. Line 43: Please give some examples of "off-label" prescribing

2. The same for line 46: "non-psychiatric diagnosis" such as: ... ?

3. The results are very well presented.

4. Regarding the discussion part, I find it very interesting but difficult to be read. English editing is required and also I would advise it to be more structured and concise. Moreover, in the first part of the discussion section, there are phrases/statements that require references. (ex: Lines 224 - 228; Line 236 and so on ...)

Author Response

I find the present article to be of extreme important for the scientific filed.

Here are some suggestions for improvement:

  1. Line 43: Please give some examples of "off-label" prescribing

Response: Thank you for kind words and all suggestions. We added a comment:

“Some examples are schizophrenia and depression, in which BZDs are often used as continuous therapy. Recommendations are that in these conditions BZDs are used only in case of agitation.”

  1. The same for line 46: "non-psychiatric diagnosis" such as: ... ?

Response: We added a comment:

“…such as musculoskeletal diseases, followed by epilepsy, hypertension, arrhythmias and chest pain.”

  1. The results are very well presented.

Response: Thank you for kind words.

  1. Regarding the discussion part, I find it very interesting but difficult to be read. English editing is required and also I would advise it to be more structured and concise. Moreover, in the first part of the discussion section, there are phrases/statements that require references. (ex: Lines 224 - 228; Line 236 and so on ...)

Response: Thank you for your suggestions. We made an effort to clear some statements and rearrange the discussion. An English proofreader was included. Regarding your comment, that references are needed in some statements – those statements are our opinion and only possible explanation of the results. We do not state that as a fact, only as opinion. We corrected and precise that in the discussion.

You can find our corrected manuscript in the attachment.

Reviewer 2 Report

The manuscript presents interesting studies on the therapeutic use of benzodiazepines. When reading the manuscript, two issues seemed worthy of consideration.

1) The authors analyzed data for 2014-2021, not distinguishing between pre-pandemic data and post-pandemic data. The outbreak of the COVID-19 pandemic in December 2019 had a significant impact on patients' mental health and the use of antidepressants, including benzodiazepines. Did the data indicate the change in the trend of benzodiazepine assignments during the pandemic?

2) How does the use of benzodiazepine in the Republic of Srpska compare to data from other countries with similar demographics?

Author Response

  • The authors analyzed data for 2014-2021, not distinguishing between pre-pandemic data and post-pandemic data. The outbreak of the COVID-19 pandemic in December 2019 had a significant impact on patients' mental health and the use of antidepressants, including benzodiazepines. Did the data indicate the change in the trend of benzodiazepine assignments during the pandemic?

Response: Thank you for your comment and suggestion. You are right, COVID-19 did exacerbate many well-controlled and triggered a few new patients with anxio-depressive disorders. Analysing the data, we noticed a slight increase in the prescription of BZD during pandemic, but it was not significant. Given that the study did not analyse the prescribing trend over the years, as well as the diagnoses on the basis of which BZDs were prescribed, we omitted commenting on the use of BZDs before and during the pandemic.

  • How does the use of benzodiazepine in the Republic of Srpska compare to data from other countries with similar demographics?

Response: We added a comments in the discussion. A mentioned studies analysed prescribing patterns in the neighbouring countries, with similar demographics as ours.

“Study by Maric et al. found that at discharge from hospital, in Croatia, Macedonia and Serbia, 81.9% of patients had benzodiazepines prescribed. They have found that factors associated with the prescriptions were exclusively clinical factors while sociodemographic factors were not found to influence benzodiazepine discharge prescriptions.”

“Study by Tahiri et al. found in Kosovo correlation between inappropriate BZD use and older age, middle education and rural environment. “

You can find our corrected manuscript in the attachment.
